# The social and environmental complexities of extracting energy transition metals

Éléonore Lèbre [1✉], Martin Stringer [2✉], Kamila Svobodova [1], John R. Owen [1], Deanna Kemp [1], Claire Côte [3], Andrea Arratia-Solar [1] & Rick K. Valenta [2]

Environmental, social and governance pressures should feature in future scenario planning about the transition to a low carbon future. As low-carbon energy technologies advance, markets are driving demand for energy transition metals. Increased extraction rates will augment the stress placed on people and the environment in extractive locations. To quantify this stress, we develop a set of global composite environmental, social and governance indicators, and examine mining projects across 20 metal commodities to identify the co-occurrence of environmental, social and governance risk factors. Our findings show that 84% of platinum resources and 70% of cobalt resources are located in high-risk contexts. Reflecting heightened demand, major metals like iron and copper are set to disturb more land. Jurisdictions extracting energy transition metals in low-risk contexts are positioned to develop and maintain safeguards against mining-related social and environmental risk factors.

[1] Centre for Social Responsibility in Mining, Sustainable Minerals Institute, The University of Queensland, Saint Lucia, QLD 4072, Australia. [2] W.H. Bryan Mining & Geology Research Centre, Sustainable Minerals Institute, The University of Queensland, Saint Lucia, QLD 4072, Australia. [3] Centre for Water in the Minerals Industry, Sustainable Minerals Institute, The University of Queensland, Saint Lucia, QLD 4072, Australia. ✉email: e.lebre@uq.edu.au; m.stringer@uq.edu.au

Climate change is reshaping the mineral resource investment landscape[1]. Major finance institutions are divesting from thermal coal and investing in the energy transition economy[2]. Combined investments in low-carbon energy technologies have reached ~30% of global energy investments[3]. The world's electric car fleet, for instance, has been growing by over 50% every year for a decade, reaching 5.1 million in 2018, a rate that would reach an international target of 100 million by 2030[4,5]. For this growth to be sustained, the worldwide production of Lithium may need to double in the next decade[6]. In addition to rapid growth, low-carbon energy technologies generally require more metal to produce the same power output as their fossil fuel counterparts. Photovoltaic power requires up to 40 times more copper than fossil fuel combustion, and wind power up to 14 times more iron[7]. More than 20 energy transition metals (ETMs), including iron, copper, aluminium, nickel, lithium, cobalt, platinum, silver and rare earth metals, are predicted to face market pressure as the production of low-carbon energy technologies intensifies[8].

Improvements in material efficiency and recycling are not sufficient to meet the increasing demand for ETMs[9]. Demand would have to be met through significant growth in resource extraction. The social and environmental implications of the anticipated rise in ETM extraction are rarely acknowledged in energy transition scenarios. Trade-off projections typically do not differentiate between the point of extraction and the remaining supply chain (e.g., refs. [10,11]). Research that expresses concern about the implications of increased ETM extraction does so without the support of global quantitative data (e.g., ref. [9]). Discussions about risk at the source of extraction instead focus on emblematic cases of metals mined in conflict or post-conflict zones. Conflict is but one factor to consider as the world transitions to a low-carbon future and demand for ETMs increase.

Mining activities alter the host environment, and tend to exacerbate pre-existing vulnerabilities[12], especially in jurisdictions where governments are unable, or unwilling, to safeguard against severe social and environmental externalities. Mineral extraction has contributed to environmental degradation, population displacement, violent conflicts, human rights violations and other adverse impacts[13]. Managing the downside risks that accompany ETM extraction sits at the core of a just transition - a transition designed to address climate change while respecting the rights of workers and communities and protecting the environment[14,15].

This paper presents a global assessment of environmental, social and governance (ESG) complexities associated with the extraction of ETMs. It uses a methodology developed to categorise and quantify source risks, i.e., risks surrounding the point of extraction[16]. A global data set of 6888 mining projects covering 20 ETMs was analysed against seven ESG risk dimensions. Each dimension is a composite indicator built from aggregate measures available in the public domain. The geographic distribution of risk factors and their co-occurrence indicates varying levels of complexity within the contexts that host extractive activities. High-risk scores across multiple dimensions translate into a high degree of difficulty in mitigating future impact scenarios[16,17]. Depending on the spatial distribution of extractive projects, ETMs exhibit different global risk profiles.

## Results

**Risk distribution across commodities**. Figure 1 connects the ESG risk profiles for a subset of ETMs (Fig. 1a, b) with their projected demand growth (Fig. 1c) and the resulting land disturbance, approximated by the movement of ore material that would be required (Fig. 1d). Demand projections were compiled from other works[18] (see Supplementary Table 1 for complete list).

For this analysis, we use mining project records extracted from the S&P Global Market Intelligence database (S&P database)[19], a comprehensive database of mining properties (see Supplementary Table 2 for estimations of production covered by the S&P database). The selected records comprise extractive projects in pre-production and operations stages, from which the next decades of global ETM production are likely to be sourced. We use this data set to assess the ESG risks associated with the demand for low-carbon energy technologies. ESG risks are analysed across a set of nine ETMs, allowing for comparison between the profiles of these metals (see Supplementary Fig. 1 for results for the full set of 20 ETMs).

The ESG risk context is modelled using seven dimensions. These include three environmental dimensions (waste, water and conservation); three social dimensions (land uses, communities and social vulnerability); and an overarching governance dimension. ESG dimensions are a reflection of ESG risk contexts, which are localised in space and time. For a given location, at the time of analysis (2019):

(1) Waste encompasses climatic, topographic and tectonic factors that play a role in mine waste containment.
(2) Water reflects the availability of fresh water and the competition that can arise between the mining industry and other water users to access fresh water.
(3) Conservation captures the proximity to key biodiversity hot spots.
(4) Land uses indicate the presence of competing land uses and associated livelihoods including agriculture and forestry.
(5) Communities relate to the presence of peoples who would be (or have been) both directly and indirectly impacted by mining operations.
(6) Social vulnerability reflects national and regional socioeconomic factors of vulnerability such as poverty, inequalities and demographic imbalance.
(7) Governance characterises the adequacy of national political and regulatory institutions.

'Methods' and Supplementary information describe the design and analysis of the sample using these risk dimensions.

Cobalt, rare earths, lithium, platinum and nickel are predicted to experience very high relative increase in annual demand (see Fig. 1c). Such high relative increases imply transformational changes for their respective sectors. For cobalt and lithium, future demand is correlated to expected production of commercial lithium-ion batteries[20]. Exponential growth in the exploration and extraction of lithium and cobalt[21] brings new risks to new locations. These two ETMs exhibit contrasting ESG risk profiles. Seventy per cent of cobalt resources by tonnage are located in contexts with high to very high ESG scores, while 65% of lithium resources are located in the very low to medium range. The two metals also differ on which risks contribute the most to the total score. Environmental risks, and particularly water, are higher for lithium, with 65% of lithium resources located in areas of medium to very high water risk, whereas social risks are higher for cobalt. The degree to which ESG risks co-occur in mining contexts is significantly higher for cobalt than it is for lithium. Ninety-eight per cent of cobalt resources with high social risks also have a high governance risk. In contrast, 53% of lithium resources located in high environmental risk contexts are also located in countries with high governance risks.

Because lithium and cobalt are almost solely used in low-carbon energy technologies, mines extracting these two metals will be of strategic importance in the energy transition. Company or government decisions to prioritise mining developments in low-risk contexts could contribute to temporarily lowering their overall commodity risk profile. However, with anticipated market

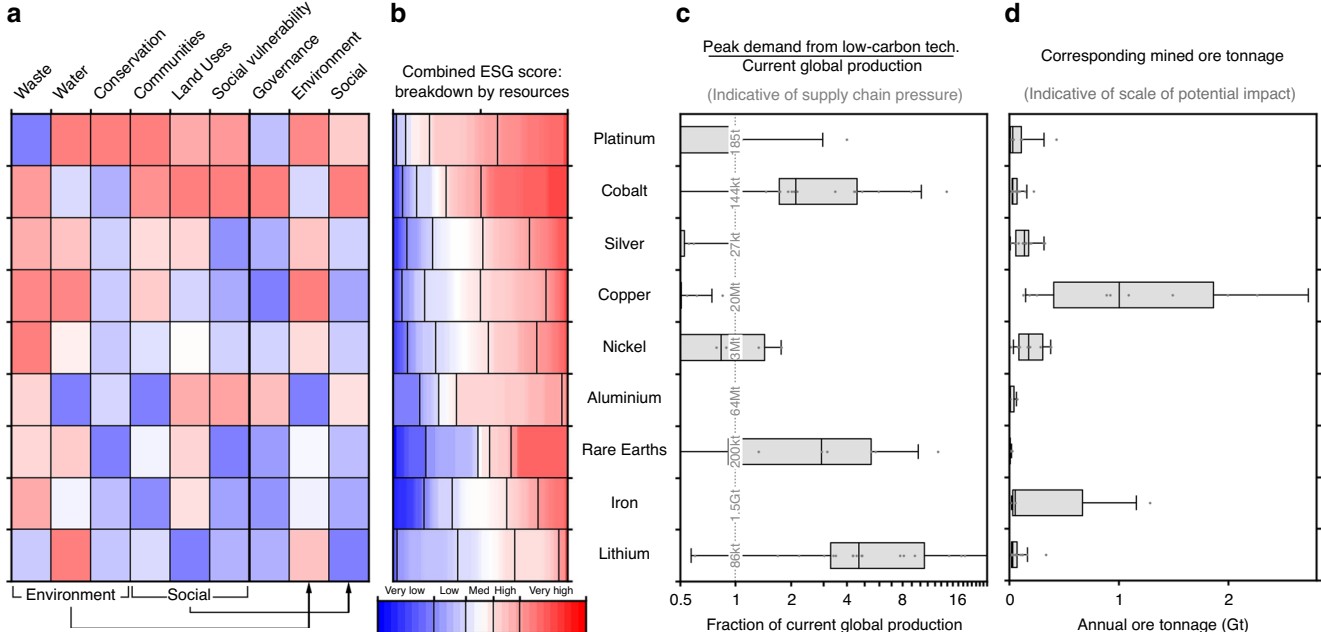

**Fig. 1 Risk profiles and demand projections for nine energy transition metals. a** Environmental, social and governance risk matrix for nine metals ranked by total score, defined as the sum of the scores for the seven dimensions (first seven columns). **b** The breakdown of total risk scores by resource tonnage. Colours correspond to risk levels. **c** The projected demand associated with growth in low-carbon energy technologies for the same nine metals, as a fraction of current global production. Individual data points and boxes that show the mean (vertical bar) and interquartile range of literature estimates ($n = 17$), and error bars show the 5th and 95th percentiles. The dashed vertical line indicates current production, with the adopted values for each metal given in small text along this line. **d** The approximate mined ore tonnage that this demand will translate to. Individual data points and boxes that show the mean (vertical bar) and interquartile range of literature estimates ($n = 17$), and error bars show the 5th and 95th percentiles.

pressure, this may only delay project development in high-risk contexts. For cobalt, a delay strategy is limited given the small number of projects in low-risk contexts. Strategies to avoid high-risk contexts may push cobalt extraction into areas where ESG risks and implications are disputed, e.g., seabed mining[22,23]. The Clarion-Clipperton seabed mining zone alone contains more cobalt than the entire global terrestrial reserve base[24]. The search for alternative sources or substitute metals will need to be supported by quantitative assessments of source risks.

Other ETMs are not expected to experience such dramatic sector growth. For these ETMs, however, absolute demand is significant. For metals like iron and copper, the relative increase in transition-related demand is small because it adds to strong demand in other sectors. Their absolute demand is high because low-carbon energy technologies require significantly more iron and copper than lithium or cobalt. Production volumes are therefore much larger for these two metals (Fig. 1d). Figure 1d shows that even a small relative demand increase may still be a major concern if the required quantity of mined material can only be sourced through multiple large-scale, low grade, open cast mines, with additional land disturbance.

Figure 1d provides an estimate of the total ore tonnage associated with the transition-related demand for each metal. Ore tonnage values are estimated using average ore grades per metal, and adjusted to account for mines that extract or will extract more than one metal. The resulting value should be interpreted as an order of magnitude estimate for the expected material movements at the mine site level, this being a reliable proxy for the extent of land disturbance attributable to each metal[25]. A higher ore tonnage means more and/or larger mines, and an overall higher land disturbance, which, in turn, increases the likelihood of land use competition, which can generate or exacerbate pressures within the surrounding social and environmental context.

Current projections, consolidated in Fig. 1d, indicate that land disturbance from the extraction of platinum could be twice that of lithium. This is because platinum grades are very low, which means large volumes of ore are required for the extraction of a few ounces of metal. Platinum has the highest overall risk score of all metals analysed in this study, with 84% of resources located in high or very high ESG risk contexts. For platinum projects, two risk dimensions prevail, social vulnerability and conservation, with both dimensions found concurrently in around 89% of platinum resources. Under these circumstances, platinum producers may be pressured to demonstrate a positive contribution to socio-economic development and nature conservation, e.g., via offset strategies.

The potential increase in land disturbance for the extraction of copper, iron and nickel is markedly higher than for lithium and cobalt. For these metals, which have a long history of use, a global energy transition will add to an already high demand for other applications. While the growth in mining activity associated with low-carbon energy technologies may be marginal compared to other uses, it will reinforce existing ESG risk conditions for these commodities, and at a much larger scale than lithium mining. For copper, iron and nickel, which have an even spread of projects in low-risk and high-risk contexts, innovation in the management and mitigation of ESG risks is critical. These findings suggest that a global extraction strategy across these sectors is needed. Sites in low complexity contexts (i.e., with a low total ESG score) where one or two high risks are present can help identify solutions to individual risks that are implementable across the sector.

**Geographic distribution of risks**. The world overview of aggregated social and environmental risk conditions shows that mining development affects regions unevenly (Fig. 2a). Hot spots are regions where higher environmental and social risks accumulate, while cold spots show regions with mining activity that are considered to be at lower risk. Although cold spots correspond to

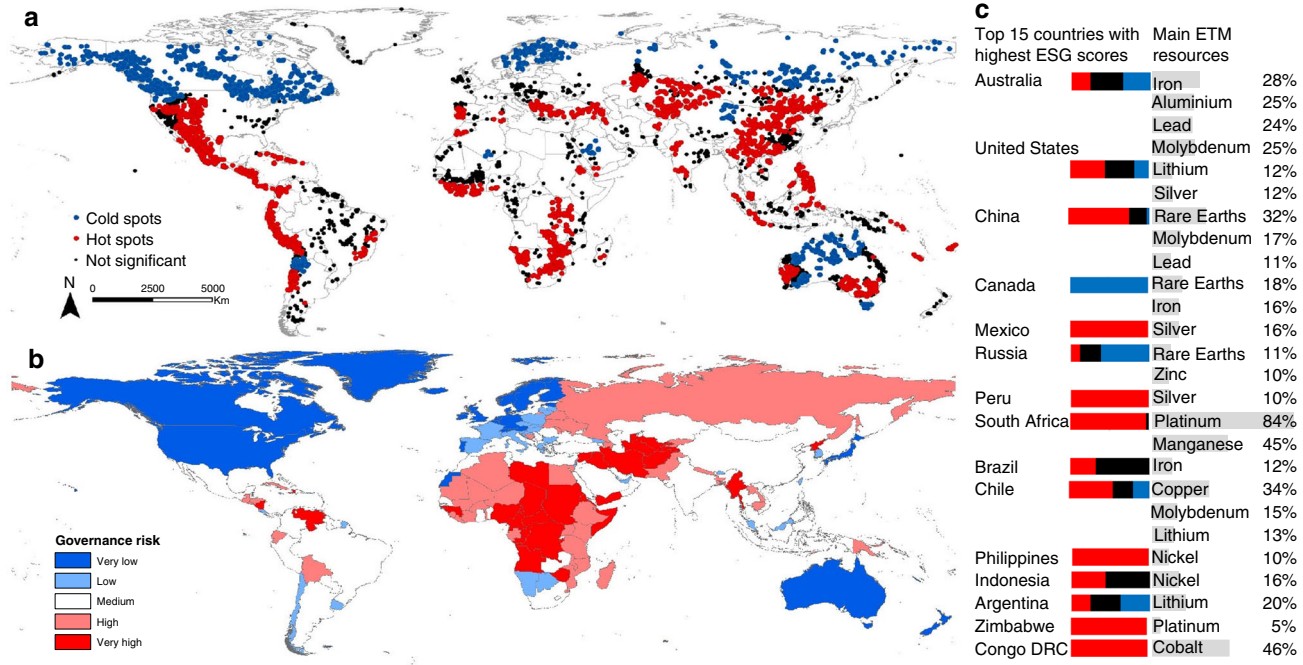

**Fig. 2 Geographic distribution of environmental social and governance risks. a** Global map of social and environmental hot spots and cold spots. Clusters of significantly low environmental and social scores (i.e., cold spots) and clusters of significantly high environmental and social scores (i.e., hot spots) are determined according to the sum of the six environmental and social dimensions. Mining projects outside hot spots and cold spots appear as black dots. **b** Global governance map. Source: Worldwide Governance Indicators, World Bank. **c** Top 15 countries according to the sum of individual mining projects' risk scores and the percentage of main energy transition metal resources in each country. Coloured bars indicate the dominance of hot spots or cold spots in the country.

lower risks, they indicate high concentrations of mining projects, which can represent a cumulative risk at regional level. The hot and cold spot distributions of individual risk dimensions are provided in Supplementary Fig. 2. Globally, there is no correlation between hot spots (Fig. 2a) and the countries' governance quality (Fig. 2b). However, 11 of the top 15 countries (Fig. 2c) also have average or above average governance risk.

The countries that yield the top 15 ESG scores (Fig. 2c) are characterised by a combination of two factors: a large number of mining projects and mining projects with high ESG scores. These countries face cumulative risk conditions and are economically reliant on the extraction of ETMs. Furthermore, these countries contribute significantly to the future production of certain ETMs, with percentages ranging from 5 to 84% of future supply, according to the data set's records.

Cold spot countries like Canada still yield a total ESG score in the top 15 due to their large number of mining projects. Canada's mining projects are located in contexts with, on average, a very low ESG risk score, i.e., fewer concurrent risk conditions. Canada, however, has the second highest number of projects in the world (1068 mining properties) after Australia (1070 mining properties). Australia and the United States, by comparison, combine a large number of mines with, on average, medium scores against the risk dimensions. Because Canada, Australia and the United States are countries with good governance scores, they may have capacity to develop and maintain safeguards against mining-related social and environmental impacts, and are well positioned to identify solutions for existing ESG risks in contexts of relative complexity. National-level assessments of source risks can help governments build capacity in the management and mitigation of these risks.

Countries with the greatest number of hot spots are China, Mexico, Peru and South Africa, totalling 575, 270, 211 and 191 mining properties, respectively, in, on average, high complexity settings. These are countries with weak or below

average governance scores. Future ETM demand is likely to drive mining developments in these resource-rich countries, placing pressure on existing social and environmental contexts. They host important proportions of key ETMs like platinum (84%), manganese (45%) and rare earths (32%). Countries with very high complexity host few mining properties and exploit few resources, exhibiting that extreme levels of ESG risks can constrain mining development. This is the case for Afghanistan, Eritrea, Ethiopia, Haiti, Uganda and Yemen.

## Discussion

The World Bank's Climate Smart Mining Facility promotes the use of low-carbon energy technologies in mining[26]. International non-governmental organisations objected to the World Bank's facility, arguing that using clean energy sources does not prevent miners from perpetuating environmental and social harm[27]. This debate highlights the dual role of the mining industry as both a negative impactor and a supplier of ETMs that are crucial for climate change mitigation. The mining industry is an intensive energy user and greenhouse gas emitter[24] and is perceived as a dirty activity that has caused adverse social and environmental impacts. The synergies and trade-offs at the source of ETM supply chains should be interrogated with greater focus and depth than has occurred to date.

The large-scale deployment of low-carbon energy technologies will continue to drive social and environmental risk. These risks can be identified by location or commodity. A global assessment of the ESG risks associated with the extraction of ETMs reveals the location of risk conditions and their combination. Identifying hot spots and locations with particular combinations of ESG risks may prompt governments, investors and other institutional actors to address acute forms of risk. Likewise, identifying ESG risks by commodity highlights the complexities and potential constraints

attached to the global supply of particular metals. Access to fresh water, for instance, is a key constraint for lithium extraction. Developing water-efficient methods in the extraction and processing of lithium could offset water scarcity issues. A high ESG score across an entire commodity, like cobalt, raises concerns about the risks of increasing its supply to meet demand.

The anticipated increase in future extractive activity comes atop a century of exponential growth in metal production. Previous mismanagement of ESG risks has created social and environmental pressures within mineral resource-rich regions[28]. This has arguably led to increased opposition to mining and resource extraction. Future ETM production faces a dual pressure: increased demand to support the transition and increased scrutiny due to adverse impacts in locations with pre-existing ESG complexity. New projects in sound governance jurisdictions will have to confirm their ability to assess, manage and minimise ESG risks, or face opposition, which may, in turn, constrain the supply of ETMs and inhibit the transition to a low-carbon future.

## Methods

**Spatial overlay**. The methodological framework consists of a spatial overlay between mining data from the S&P Global Market Intelligence database—a commercial database that gathers public disclosures from mining companies—and publicly available data sets assembled into seven ESG dimensions. This approach, used in previous work[16,17,29–31], models the interface between a mining project and the geographic context in which it is located. It focuses on building an understanding of inherent or latent complexities present in the external context. The industry's engagement and management of risks in different contexts determine whether these risks are static, exacerbated or reduced. We assume that a high score in any ESG dimension drives up both the likelihood and the severity of the consequences of a detrimental event (e.g., tailings dam failure, road blockade) occurring at the interface between the mine and its context and potentially having consequences to the developer, local people and the environment.

The S&P database includes records of operating mines, as well as mining projects in pre-production stages including the target outline stage: advanced exploration, prefeasibility and scoping, feasibility, construction and commissioning stages. Records of early stages grassroots and exploration projects were excluded on the basis that their future development is too preliminary, meaning any estimation of contained resources would be unreliable. For each mining project included in the analysis, we extracted the spatial coordinates and the most up to date resources estimates. Resources estimates represent current known metal content and discount material already extracted. Due to reporting norms, these estimates are likely to be understated. Data are current as of May 2019.

Conducting a global review at resource-scale inevitably involves some constraints on data. The S&P database relies on public disclosure, and the level of disclosure varies according to commodities and countries. For the nine ETMs presented in this paper, the S&P database covers between 72% for aluminium and 100% for platinum of current global production (see Supplementary Table 2). Noting the S&P database does not provide a coverage estimate for rare earths. Resource coverage for pre-production projects in the S&P database is also unknown. Nonetheless, the S&P database remains one of the best available repositories for mining data.

The seven ESG dimensions were constructed using the methodology on composite indicators from the Organisation for Economic Co-operation and Development[32]. The first step is the review and selection of global variables that serve as proxies for different ESG aspects and that can be companied together into ESG dimensions. Constraints for these proxies to accurately depict the ESG mining context include the availability of global data and the quality of this data (resolution levels, completeness and methodological choices made by the authors of the data sets). The 6888 mining projects were overlain with each variable and attributed location-specific values. Overall, 24 global variables from 14 different sources were applied. Of those 24 variables, 8 are national-level indexes and 16 are rasters and polygons data sets that were overlayed to the 6888 point data set on ArcGIS. The theoretical framework and data selection are summarised below, and visualised in Supplementary Fig. 3. Supplementary Data 1 provides source, download links and description for all variables. Correlations across ESG dimensions and across the 24 variables are presented in Supplementary Tables 3 and 4, respectively. The contribution of each variable to the overall risk dimensions is visualised in Supplementary Figs. 4–10. Other methodological steps including data aggregation, normalisation and weighing are detailed in Supplementary Tables 5–7. The robustness of our methodology is tested in Supplementary Figs. 11 and 12.

**Waste**. Mining projects are characterised by large material movements that occur throughout the mine's life cycle. These movements result in waste stocks and mining voids, which are the main cause of direct land disturbance. In terms of land use, tailings storage facilities can cover half of the area of disturbance[33]. Waste rock

dumps and voids, including open pits and underground workings, cover most of the remaining land. Mine owners are responsible for minimising the impacts of their activities on the host environment by rehabilitating disturbed areas and ensuring the effective containment or neutralisation of polluting substances.

Natural conditions in and around mine sites pose challenges to the design, construction and maintenance of waste facilities and mining voids. Reactive substances within the unearthed material and void walls are exposed to wind, rain and oxygen, which favour their reaction and the diffusion of pollution through either dust or acid drainage. Miners have to plan for long-term containment and ensure the structural integrity of waste facilities. In extreme cases, a tailings dam failure can cause major impacts to local communities and ecosystems. The causes of these failures are often multiple and include human and management error as well as external factors such as heavy rains and earthquakes[33,34]. The waste dimension takes into account both the risk of catastrophic failures and of chronic seepage and airborne pollution. The risk indicators that were selected to build this ESG dimension included seismic risk, cyclones risk, average wind speed and maximum annual precipitation. A fifth indicator represents terrain ruggedness, which is a topographic factor that expresses the variability of elevation in an area, and adds complexity to the construction of large and stable structures. Further explanation on the waste dimension is provided in Supplementary Note 1.

**Water**. Mining and mineral processing activities at mine sites usually have high fresh water requirements[35]. Fresh water here refers to high-quality water which is suitable for human consumption or would require limited treatment to make it suitable for human consumption. Access to fresh water can be a challenge in contexts of water scarcity and/or competing water uses[29]. Inadequate mine water management involving high withdrawal, low rates of water reuse and discharge of contaminated water can heavily impact local water resources and affect surrounding ecosystems and communities. The water dimension only quantifies the risk of not securing sufficient access to fresh water. The risk of discharge is partially covered in the waste dimension. There are other factors, such as the sensitivity of receiving environment, local regulatory frameworks and associated water quality objectives, that contribute to the risk of discharge but cannot be assessed at the selected global scale.

In terms of access to fresh water, the World Resources Institute's provides indicators of water supply risk relevant to mining[36]. The Baseline Water Stress indicator and the Inter-annual Variability indicator were selected for their level of completeness and their complementarity in illustrating water supply risk, as they account for two main factors contributing to securing access to fresh water: persistent low fresh water availability and significant variations in fresh water availability with time.

**Conservation**. Extractive activities affect natural habitats both within and outside the mining lease. Mining infrastructure built to access and transport the ore creates large corridors that expand the disturbance beyond the mining area. The risks generated by the proximity between mines and critical biodiversity preservation areas have been flagged multiple times (e.g., refs. [12,31,37]). The great majority of the planet's biodiversity is not currently under strict legal protection[38], meaning it is potentially exposed to mining development and other human land uses. For this dimension, we use three nature conservation spatial data sets, the Key Biodiversity Areas, hosted by Birdlife International[39], the Biodiversity Hotspots map by the Critical Ecosystem Partnership Fund[40] and the Total Species Richness maps provided by Jenkins et al.[41]. The three data sets use different definitions of conservation priorities and complement each other. The first two are polygon data sets, and the measure used for them is the distance from mining project points to the closest polygon. The Total Species Richness data sets are raster data sets that provide further granularity on the distribution of centres of richness for vertebrate species.

**Communities**. People living or working in the vicinity of a mining project are the key stakeholders and bearers of social risk. People who were present before the mine's development have had to make way and adapt to mining-induced social, economic and environmental changes from exploration to operation phase to closure and post-closure. In-migration of workers and artisanal miners[42] and displacement and resettlement of populations[43] are examples of practices and social phenomena with complex consequences. We use the European Commission's Global Human Settlement Layer, which provides population density raster with 1 km resolution, to assess the population density value in direct proximity to the mining project[44]. This provides an indication of the presence of directly affected communities. In addition, we assess population density in a 100 km buffer around the mine to account for indirect or chronic impacts in the wider area of influence.

Some social groups are affected more than others. Indigenous peoples often experience higher levels of poverty, marginalisation and discrimination, while maintaining deep spiritual, cultural and sometimes legal ties to their land[45]. The location of a mine on indigenous land adds a degree of complexity to the social context and involves additional risks during the land access and acquisition process and throughout project expansion. We therefore complement the communities dimension with the Indigenous Peoples Map developed by Garnett et al.[46].

**Land uses**. Constrained access to land and management and stewardship of land are the main risks faced by the mining industry[47]. Extractive activities are bound to take place where the orebody is situated, and rearrange existing land uses. Competition between mining and other human land uses is anticipated to increase as population growth continues[38]. Mining development and agricultural activities also tend to progressively expand, often into forestland, compromising environmental assets and threatening the livelihoods of people reliant on natural resources[12]. Mining infrastructure such as road and rail networks can enable population movements and expands the mine footprint far beyond the mining lease[48]. Global cropland and pastureland layers built from satellite imagery were selected from NASA's Socioeconomic Data and Applications Centre[49,50], complemented with the forest extent map from the Japan Aerospace Exploration Agency[51]. These three layers indicate the presence of farmland and forestland in the vicinity of the mining project, which would be then likely to compete against existing livelihoods.

**Social vulnerability**. A mine and its local context are situated in a wider social context, which presents varying levels of vulnerability at different scales—local, regional and national. Vulnerability is the propensity or predisposition of an individual or group of people to suffer damage and loss, including loss of life, livelihood and property or other assets. Vulnerability to external stressors like natural or man-made disasters is, in part, a social condition[52], and the presence of a mine constitutes the element of exposure to potential acute or chronic issues, testing societal resilience. A variety of factors contributes to social vulnerability, namely, the susceptibility of groups and individuals to harm and their ability to respond and mitigate that harm. We reviewed relevant social data available at global scale and performed a principal components analysis to select three main uncorrelated indicators: the United Nations Development Programme's Human Development Index, the World Bank's Gini coefficient, and the Total Dependency Ratio compiled by NASA Socioeconomic Data and Applications Centre. Together, the three indicators and indexes combine societal, household and individual level dynamics. Health, education and poverty for the Human Development Index, income inequalities for the Gini coefficient and age dependence and family structure for the Total Dependency Ratio are identified as key factors in understanding social vulnerability[53].

**Governance**. Finally, the governance of a country influences both the mining project and its social and environmental context. Robust governance frameworks support the fair redistribution of mining revenues, the protection of citizens and the environment and a clear and consistent permitting and approval process for the major project developments. Poor governance provides a permissive environment for suboptimal performance from the operator, and fuels inequalities, grievance and distrust within local populations. Poor governance contributes to a climate of vulnerability and tension, potentially leading to production disruption[54] and social unrest[55]. For this dimension, we selected the Worldwide Governance Indicators developed by the World Bank[56], which cover all aspects of a country's governance relevant to mining. They include the robustness of regulations and policies, the effectiveness of public services and the degree to which rules like property rights are enforced. They also account for social and political stability, including perceived corruption of power, and the respect of freedom and human rights.

**Environmental social and governance risk matrix**. To build the ESG risk matrix in Fig. 1a, we took the resource-weighted average of individual mining projects, using the formula:

$$\text{Risk}_x \text{ of Metal}_y = \frac{\sum_{i=1}^{6888} \text{Resources}_{i,y} \text{Risk}_{i,x}}{\sum_{i=1}^{6888} \text{Resources}_{i,y}},$$

where resources are expressed in tonnes, ounces or pounds depending on the metal (see Supplementary Table 8).

**Transition-related demand and ore tonnage forecasts**. The demand forecasts for selected metals shown in Fig. 1b were consolidated from the results of 17 separate studies, as shown and listed in Supplementary Table 1. Demand estimates were converted to a fraction of current production, using the average of the 2018 and 2019 production values listed in the US Geological Survey's Mineral commodity summaries[57].

The demand for rare earths refers to the demand for rare earth oxide ore, which was estimated from the forecast demand for the constituent metal oxides using the relative composition of the world's major deposits[58], and production from these deposits (S&P database, USGS 2020). According to these sources, each tonne of rare earth oxide produced corresponds to an average of about 12 kg dysprosium and 180 kg neodymium production, the two rare earth metals that are consistently forecast to experience high demand due to the energy transition. The demand for neodymium is typically forecast to be higher than for dysprosium (ratios of about 6:1), but the yield for dysprosium is so much lower (a ratio of about 1:15) that the demand for dysprosium would be the driving factor for rare earth production in these forecast scenarios. The demand for rare earths in Fig. 1c is therefore set to the relevant forecast demand for dysprosium, divided by 0.012.

To translate the demand to an approximate ore tonnage, we divided these by an effective grade value calculated from the grades and production reported in the S&P database (see Supplementary Fig. 13). This effective grade is the total worldwide production of the metal commodity divided by the total ore tonnage mined worldwide to produce it (noting that for bauxite projects, the S&P database displays grades as alumina percentages, which we translated into aluminium grades). To avoid double-counting ore tonnage for mines that produce multiple commodities, the ore tonnage corresponding to a particular commodity is weighted by the value of that commodity's production as follows:

$$\text{Effective grade} = \frac{\sum_{\text{mines}} \text{Production}}{\sum_{\text{mines}} \left( \frac{\text{Production}}{\text{Grade}} \times \text{Value share} \right)},$$

where the summation is over all mines in the database that report grades for that commodity. The value share of a given commodity in a given mine is defined as the:

$$\text{Value share} = \frac{\text{Production} \times \text{Price}}{\sum_{\text{commodities}} \text{Production} \times \text{Price}},$$

where the summation is over all metal commodities produced at that mine site. The price used for the purpose of this estimate was the 2019 average published on the S&P database.

**Geographic distribution of hot spots and cold spots**. Figure 2a shows the geographic distribution of statistically significant hot spots, i.e. clusters of high environmental and social scores (in red), and cold spots, i.e. clusters of low total environmental and social scores (in blue). Statistically significant hot spots and cold spots have high scores and at the same time are surrounded by other projects with high scores. Insignificant spots (in black) are not part of any statistically significant cluster. The significance of hot/cold spots was measured at a confidence level of 90%. This was done using the statistical tool Optimized Hot Spot Analysis in the Spatial Statistics set of ArcGIS 10, which analyses each mining project within the context of its neighbouring projects. Supplementary Fig. 14 shows how hot and cold spots vary for different metal subgroups.

The top 15 country list in Fig. 2c is determined from the sum of total ESG scores of all individual mining projects in a given country. Supplementary Table 9 shows how the top 15 list varies for different metal subgroups. Supplementary Tables 10 and 11 provide further details on top hot spot and top cold spot countries, respectively.

**Reporting summary**. Further information on research design is available in the Nature Research Reporting Summary linked to this article.

## Data availability

Download links and description of all publicly available data sets used for this study are provided in Supplementary Data 1. Due to their proprietary nature, mining project data are only available from the corresponding author upon reasonable request. Source data are provided with this paper.

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

## Acknowledgements

We are grateful for the strategic funds received from The University of Queensland in support of the Sustainable Minerals Institute's cross-disciplinary research on complex orebodies. We acknowledge the organisations and people that have produced the data sets we used in our analysis. Particular thanks to S. Garnett and colleagues, B. McLellan, T. Watari, and A. Manberger for sharing data from their work and for helpful discussions. We also acknowledge computer resources provided by the Dow Centre for Sustainable Engineering Innovation.

## Author contributions

The initial design of this research project originates from R.K.V.; methodological development and analysis for this paper were conducted by É.L., J.R.O., K.S. and M.S.; D.K., C.C., R.K.V. and A.A.S. made contributions on methodological choices related to their areas of expertise; É.L. and M.S. designed Fig. 1; É.L. and K.S. designed Fig. 2; É.L. drafted the first version of the paper; all authors provided revisions and additions in the subsequent draft versions.

## Competing interests

The authors declare no competing interests.
