## [Peer Review File · Nature Communications]

REVIEWER COMMENTS

Reviewer #1 (Remarks to the Author):

This paper addresses the social and environmental impacts of global mining for metals that will be needed for a transition of the global energy system away from fossil fuels (“energy transition metals, or ETM). There are two types of claims made. The first claim is that the proposed set of global composite environmental, social, and governance indicators are an appropriate method for identifying the social and environmental risk factors of mining. The second claim is the identified findings: that 84% of platinum and 70% of cobalt resources are in high-risk locations, that iron and copper will disturb larger land areas, and that Australia, USA and China host high concentrations of ETM mines with multiple risk factors.

This is a creative, well written and useful manuscript that characterizes the social and environmental risks of energy transition metals. This is the first and only paper to broadly characterize the social and environmental risks, globally, of all of the metals in demand for renewable energy technologies. It develops a methodology that can also be applied to other global industrial activities. The authors have developed a very large and definitive database of mining sites, both operational and under development, and have coupled this with use of the leading databases relevant governance, social vulnerability, mining risk, and conservation. The analysis includes an excellent approach to conservation risk identification, through combination of the data on Key Biodiversity Areas, Biodiversity Hotspots, and Total Species Richness (Line 387), and an excellent approach to evaluation of social vulnerability, through combination of the human development index (HDI), the total dependence ratio (TDR), and GINI coefficient for inequality (Line 434), and excellent characterization of governance risk through use of the Worldwide Governance Indicators (Line 451).

The manuscript could be strengthened with further effort on both of the claims (methodology and findings) mentioned above.

In terms of the appropriateness and robustness of the method, a thoughtful sensitivity analysis or robustness evaluation is needed. There are three dimensions of the open questions regarding the robustness of the method: (a) with a different selection of indicator databases and implementation procedures, how would the results change; (b) with different variations in the details of the method, which countries and which metals see the most change in their evaluation; (c) with different selection of metals (see below) how would the results change. It is important to evaluate whether through the specific choice of metals or choice of combinations and weighting of indicators, if some any of the countries highlighted as hot spots or cold spots might “flip”, or become neutral, or disappear from the analysis.

In terms of the specific findings from applying the method:

- The statement of findings in the abstract is “that 84% of platinum and 70% of cobalt resources are in high-risk locations, that iron and copper will disturb larger land areas, and that Australia, USA and China host high concentrations of ETM mines with multiple risk factors”: These findings do not sufficiently highlight the features of the method and the findings that are provided; more interesting findings might be extracted. The method identifies hot spots, cold spots, and neutral spots. The cold spots and neutral spots findings are the most novel; it is very common to read that mining has significant environmental and social costs. A braver and more novel abstract and result could say something like: “84% of platinum and 70% of cobalt resources are in high-risk locations, while nearly half of rare earths and lithium resources are in low-risk locations” (see Figure 2c).
- The statement in the abstract that “iron and copper will disturb larger land areas”, while surely true in general, is not core to the question of metal demand for energy transitions: Figure 1b shows that the increased demand for iron and copper (and silver and aluminum) for the energy

transitions will be zero. Figure 1c shows large amounts of iron and copper being mined; this is because they are major metals and their quantities dwarf the production of the other metals discussed. It is surprising that iron and copper are included in this study; they are apparently mentioned in reference 8, but even the authors acknowledge, on line 205, that the increase in mining of copper, iron and nickel due to the energy transition will be marginal (lines 205-214). This finding about iron and copper disturbing larger land areas should be dropped from the abstract; perhaps these and other major metals that will not see major demand change due to energy transitions should all be dropped.

- If iron, copper, silver, nickel, aluminum and lead were removed from the analysis, how would the results change? These are not the major metals typically considered as growth metals for new energy technologies. Iron, aluminum and lead are the three main ETM resources listed in Figure 2c for Australia; would Australia's prominence be reduced if the metals considered were restricted to those that will see significant change with development of energy transition technologies?
- Regarding the statement in the abstract that "Australia, USA and China host high concentrations of ETM mines with multiple risk factors": rather than "high concentrations of ETM mines" it should be something like "high production of ETMs."
- The most intriguing results of the paper is the identification of "cold spots" in terms of environmental and social impacts of mining. Highlight these! How about a statement in the abstract something like "mines in Canada, Alaska, northern Chile, Scandinavia, Russia, Saudi Arabia, southern Algeria, and most of Australia have very low environmental and social impact risks." These are novel findings.

Supporting Information

Section 2.2.2 Social Vulnerability

On line 434 of the main text, it is stated that the three indicators: HDI, TDR, and GINI, are uncorrelated and will combine national (GINI), household (TDR) and individual (HDI) levels. However, now in the supporting information, it is stated that the HDI is being given three times the weight of either the TDR or the GINI. I do see that that HDI has three measures – health, education and income – and that these 3 HDI measures are being weighted equally with the GINI and TDR. However, some justification is needed. Perhaps the thought is that the individual (HDI) measures are more important than household or national levels. Or perhaps the thought is that the HDI measures are more robust than the TDR or GINI. Or perhaps there isn't a solid reason for choosing this weighting rather than something else. Perhaps the best approach is to state that there is no basis for any particular weighting, and then show how the results are different if only the HDI is used; if only the TDR is used, and if only the GINI is used. They should be quite different, given that it was stated that the measures are uncorrelated. By showing a sensitivity analysis, readers can avoid over-interpreting the results that may be a function of the specific weightings chosen.

Section 2.2.7 Waste.

More support for, or discussion of, the relationship between these waste indicators and waste impact events would be helpful. Wind speed, for example, can be high in areas that are otherwise very stable. There is a plausible connection between waste incidents and at least some of these indicators. Even so, this is a part of the analysis that appears less well supported.

Figure S17 clearly illustrates how different risk dimensions vary geographically. What would also be helpful is a version of Figure 2c, showing either how that figure changes for different metals choices, risk dimensions, or weightings.

- Valerie M. Thomas

Reviewer #2 (Remarks to the Author):

This paper presents a novel approach to categorizing environmental and social risk in meeting future demand of technology critical metals. The authors have used a large data set of mining projects in operation and close to development up to May 2019 to consider the overlap in 7 key dimensions of risk, derived in turn from 21 indicators which have been aggregated using the OECD composite indicators methodology (the supplementary material admirably presents a correlation analysis for the indicators as well). A point which deserves clarification in the methodology section is how georeferencing of the mine sites was linked to the indicators as it is unclear if each of those indicators has geographically specific data. The spatial resolution for the indicators data that is being linked to the mine site needs clarification as often such data is available only at the national level.

Overall this paper is an important contribution to the literature because it presents a systems approach to considering risk in socio-ecological terms. The simple categorization of hotspots, cold spots and neutral black spots has easy policy evaluation appeal. An intriguing finding of the paper is that governance deficiency does correlate in aggregate with hotspot prevalence, despite many hotspot countries having poor governance in the overall rankings. This shifts the conversation from mining acceptability risks being considered as a developing world problem to one which has much broader context. This paper merits publication but there are a few areas which need revision or nuanced coverage before moving forward.

a) Demand forecast: The data in Figure 1 b can be questioned in terms of how metals like aluminum are relegated to a lower priority. Indeed a new report on critical metals for carbon mitigation from the World Bank (May, 2020) notes the salience of aluminum. The authors need to revisit the demand forecasts and consider this latest analysis with greater care:
<http://pubdocs.worldbank.org/en/961711588875536384/Minerals-for-Climate-Action-The-Mineral-Intensity-of-the-Clean-Energy-Transition.pdf>

b) Static and dynamic risk categories: There is a major difference between some of the ecological risk categories and the social and governance risk categories in terms of temporal factors. The Amazon rainforest or the Congo basin has inherent biodiversity value and irreparable damage there suggests that this is a static risk which can't be diminished (hence call for permanent protected areas in some cases). On the other hand social and governance risks are dynamic and change frequently. Countries which may be stable one day can be unstable the next or vice versa. The authors need to differentiate these factors in their discussion – particularly with reference to their headline finding on platinum group metals. Given the dominance of South Africa in this regard and the dominance of Social and Governance factors in this conclusion, there should be a note about this matter. Furthermore, there is now much higher recycling of platinum from catalytic converters as a result of such risks and authors need to note this aspect as well. The authors quote Rasmussen (2019) for their demand calculations but should note the recycling aspects in that paper as well.

c) Processing impact: Surprisingly, rare earth metals come out as rather “green” in this analysis possibly because they are mined as companion metals in the larger deposits or the database does not cover the smaller mining sites in Jiangxi province of China. Furthermore, the chemical processing of rare earths has major environmental and social consequences which are also not addressed in this analysis. Some discussion of this anomaly is worth addressing. The work of Dr. Julie Klinger in this arena may be worth consulting and referencing.

d) Oceanic sources: Given the conclusion of this study that the technology metals, particularly cobalt will likely face major ESG risk, the authors should at least mention that oceanic minerals,

particularly the massive reserves of cobalt deserve to be noted more clearly than a tangential reference in citation 27. This is particularly salient since the International Seabed Authority is currently formulating regulations precisely to address these concerns and provide alternative supply scenarios.

Once these revisions are addressed, this paper would be ready for publication and a welcome contribution to the field.

Saleem H. Ali
Minerals, Materials and Society Program, University of Delaware

Response to reviewer comments

Reviewer 1:

#	Reviewer comment	Response
1	This paper addresses the social and environmental impacts of global mining for metals that will be needed for a transition of the global energy system away from fossil fuels ("energy transition metals, or ETM). There are two types of claims made. The first claim is that the proposed set of global composite environmental, social, and governance indicators are an appropriate method for identifying the social and environmental risk factors of mining. The second claim is the identified findings: that 84% of platinum and 70% of cobalt resources are in high-risk locations, that iron and copper will disturb larger land areas, and that Australia, USA and China host high concentrations of ETM mines with multiple risk factors. This is a creative, well written and useful manuscript that characterizes the social and environmental risks of energy transition metals. This is the first and only paper to broadly characterize the social and environmental risks, globally, of all of the metals in demand for renewable energy technologies. It develops a methodology that can also be applied to other global industrial activities. The authors have developed a very large and definitive database of mining sites, both operational and under development, and have coupled this with use of the leading databases relevant governance, social vulnerability, mining risk, and conservation. The analysis includes an excellent approach to conservation risk identification, through combination of the data on Key Biodiversity Areas, Biodiversity Hotspots, and Total Species Richness (Line 387), and an excellent approach to evaluation of social vulnerability, through combination of the human development index (HDI), the total dependence ratio (TDR), and GINI coefficient for inequality (Line 434), and excellent characterization of governance risk through use of the Worldwide Governance Indicators (Line 451).	We appreciate the reviewer's constructive feedback. We have strengthened our arguments as suggested by the reviewer.
2	The manuscript could be strengthened with further effort on both of the claims (methodology and findings) mentioned above. In terms of the appropriateness and robustness of the method, a thoughtful sensitivity analysis or robustness evaluation is needed. There are three dimensions of the open questions regarding the robustness of the method: (a) with a different selection of indicator databases and implementation procedures, how would the results change; (b) with different variations in the details of the	We agree that a sensitivity analysis would be helpful. We have added several sections to the Supporting Information following the reviewer's specific comments: (a) with a different selection of indicator databases and implementation procedures, how would the results change. We have added a sensitivity analysis for each of the seven ESG dimensions in section 2.2 of the supplements. The new

	method, which countries and which metals see the most change in their evaluation; (c) with different selection of metals (see below) how would the results change. It is important to evaluate whether through the specific choice of metals or choice of combinations and weighting of indicators, if some any of the countries highlighted as hot spots or cold spots might “flip”, or become neutral, or disappear from the analysis.	figures show the contribution of each indicator to the overall dimension. The figures show varying sensitivities from one metal to another. In addition, new section 2.4 presents a sensitivity analysis that tests the stability of figure 1a. (b) with different variations in the details of the method, which countries and which metals see the most change in their evaluation. We have added a section in the supplements titled “Top 15 countries for selected metal groups”, which looks into how the Top 15 countries by sum of total ESG scores vary depending on which metals are considered. (c) with different selection of metals (see below) how would the results change. We have added a section in the supplements titled: “Spatial distribution of hot spots and cold spots for selected metal groups”. This section provides additional maps of cold spots and hot spots for 5 different metal subgroups. In particular, it allows for comparison between the specialty metal group (lithium, cobalt and rare earths) identified in figure 1c, and the major metal group (iron, copper and nickel) identified in figure 1d.
3	In terms of the specific findings from applying the method: The statement of findings in the abstract is “that 84% of platinum and 70% of cobalt resources are in high-risk locations, that iron and copper will disturb larger land areas, and that Australia, USA and China host high concentrations of ETM mines with multiple risk factors”: These findings do not sufficiently highlight the features of the method and the findings that are provided; more interesting findings might be extracted. The method identifies hot spots, cold spots, and neutral spots. The cold spots and neutral spots findings are the most novel; it is very common to read that mining has significant environmental and social costs. A braver and more novel abstract and result could say something like: “84% of platinum and 70% of cobalt resources are in high-risk locations, while nearly half of rare earths and lithium resources are in low-risk locations” (see Figure 2c).	Point taken. A sentence in the abstract was changed to emphasize the cold spots findings. We are mindful of not labelling lithium as “low risk” since lithium does have one predominant risk, which is water. 65% of lithium resources are located in areas of high water risk. We added this percentage to the text to provide more clarity.
4	The statement in the abstract that “iron and copper will disturb larger land areas”, while surely true in general, is not core to the question of metal demand for energy	Figure 1b shows the relative increase in demand as a fraction of current demand. If current demand is high, as it is the case for

	transitions: Figure 1b shows that the increased demand for iron and copper (and silver and aluminum) for the energy transitions will be zero. Figure 1c shows large amounts of iron and copper being mined; this is because they are major metals and their quantities dwarf the production of the other metals discussed. It is surprising that iron and copper are included in this study; they are apparently mentioned in reference 8, but even the authors acknowledge, on line 205, that the increase in mining of copper, iron and nickel due to the energy transition will be marginal (lines 205-214). This finding about iron and copper disturbing larger land areas should be dropped from the abstract; perhaps these and other major metals that will not see major demand change due to energy transitions should all be dropped.	major metals like iron and copper, the relative increase actually corresponds to a high tonnage. In terms of absolute production tonnage, iron and copper are needed in much higher quantities than lithium or cobalt in the energy transition. To clarify this, we have amended figure 1b (now figure 1c), which now includes current global production values, so the reader can more easily visualise what a relative increase in demand represents. We have also clarified these aspects in the abstract and in the main text.
5	If iron, copper, silver, nickel, aluminum and lead were removed from the analysis, how would the results change? These are not the major metals typically considered as growth metals for new energy technologies. Iron, aluminum and lead are the three main ETM resources listed in Figure 2c for Australia; would Australia's prominence be reduced if the metals considered were restricted to those that will see significant change with development of energy transition technologies?	It is interesting to look at how the Top 15 list of countries varies depending on which ETMs are considered. We have added a section in the supplements covering this aspect. One can see that the Top 15 list does vary depending on metal choices, however many countries (Australia, USA, Canada, China, Russia, Chile, South Africa, Philippines etc.) consistently appear across the different Top 15 lists. Australia maintains a very high rank (in either first or third position) regardless of the metal selection. Regarding the inclusion of major metals in the analysis, please refer to our response to comment 4.
6	Regarding the statement in the abstract that "Australia, USA and China host high concentrations of ETM mines with multiple risk factors": rather than "high concentrations of ETM mines" it should be something like "high production of ETMs."	The sentence in the abstract was amended.
7	The most intriguing results of the paper is the identification of "cold spots" in terms of environmental and social impacts of mining. Highlight these! How about a statement in the abstract something like "mines in Canada, Alaska, northern Chile, Scandinavia, Russia, Saudi Arabia, southern Algeria, and most of Australia have very low environmental and social impact risks." These are novel findings.	We provided additional information in the abstract emphasizing the cold spots findings. Canada and Australia are the two countries that have the largest cold spots in terms of number of ETM mines.
8	Supporting Information Section 2.2.2 Social Vulnerability On line 434 of the main text, it is stated that the three indicators: HDI, TDR, and GINI, are uncorrelated and will combine national (GINI), household (TDR) and individual (HDI) levels. However, now in the supporting information,	This weighting approach to reflect the fact that the HDI, GINI and TDR are different in nature. The HDI is an index, made up of three dimensions that use their own specific indicators, which have already gone through a normalisation and

	it is stated that the HDI is being given three times the weight of either the TDR or the GINI. I do see that that HDI has three measures – health, education and income – and that these 3 HDI measures are being weighted equally with the GINI and TDR. However, some justification is needed. Perhaps the thought is that the individual (HDI) measures are more important than household or national levels. Or perhaps the thought is that the HDI measures are more robust than the TDR or GINI. Or perhaps there isn't a solid reason for choosing this weighting rather than something else. Perhaps the best approach is to state that there is no basis for any particular weighting, and then show how the results are different if only the HDI is used; if only the TDR is used, and if only the GINI is used. They should be quite different, given that it was stated that the measures are uncorrelated. By showing a sensitivity analysis, readers can avoid over-interpreting the results that may be a function of the specific weightings chosen.	aggregation process. The GINI coefficient is a statistical measure of wealth distribution, with values that currently vary between 25 and 63. The TDR is a percentage, which can range from almost zero to more than 200%. We acknowledge that all the measures chosen for this analysis are proxies of complex ESG factors and that these proxies are necessarily imperfect, due to various reasons including data availability. This is particularly the case for “qualitative” dimensions like the social and governance dimensions, but also environmental dimensions, which are subject to approximation, as the reviewer rightly pointed out in the next comment. We have added a sensitivity analysis for this dimension, in which we evaluate how each indicator contributes to the overall Social Vulnerability score (see comment 2). In introduction of section 2.2, we now acknowledge inherent limitations of our approach. We have also made a minor correction to the text in the methodology section of the manuscript, and added further details in the Social Vulnerability section of the supplements.
9	Section 2.2.7 Waste. More support for, or discussion of, the relationship between these waste indicators and waste impact events would be helpful. Wind speed, for example, can be high in areas that are otherwise very stable. There is a plausible connection between waste incidents and at least some of these indicators. Even so, this is a part of the analysis that appears less well supported.	We have added further details on indicator choices in section 2.2.7, where we discuss the relationship between waste indicators and waste impact events. We also have performed a sensitivity analysis to evaluate the effects of changes in indicators (see comment 2).
10	Figure S17 clearly illustrates how different risk dimensions vary geographically. What would also be helpful is a version of Figure 2c, showing either how that figure changes for different metals choices, risk dimensions, or weightings.	We added a new figure in the supplements titled “hot and cold spots distribution for selected metal groups”. This figure shows different versions of Figure 2c for different metal choices. One can see that the main hot spots and cold spots remain the same regardless of the metals selected. This is because this analysis relies on two factors: 1) the sum of environmental and social risk scores at each individual mine site, and 2) the distance between sites. Removing certain metals from the mix only affects the latter, and results in a lower density of mining projects and therefore fewer hot

		and cold spots. We addressed the considerations on risk dimensions and weightings in the sensitivity analysis added to sections 2.2 and 2.4 of the supplements, see comment 2.
--	--	---

Reviewer 2:

#	Reviewer comment	Response
11	This paper presents a novel approach to categorizing environmental and social risk in meeting future demand of technology critical metals. The authors have used a large data set of mining projects in operation and close to development up to May 2019 to consider the overlap in 7 key dimensions of risk, derived in turn from 21 indicators which have been aggregated using the OECD composite indicators methodology (the supplementary material admirably presents a correlation analysis for the indicators as well). A point which deserves clarification in the methodology section is how georeferencing of the mine sites was linked to the indicators as it is unclear of each of those indicators has geographically specific data. The spatial resolution for the indicators data that is being linked to the mine site needs clarification as often such data is available only at the national level.	A table with complete list of indicators is now available in the data package uploaded to Figshare. The table includes resolution levels. We also added clarification on this aspect in the methodology section.
12	Overall this paper is an important contribution to the literature because it presents a systems approach to considering risk in socio-ecological terms. The simple categorization of hotspots, cold spots and neutral black spots” has easy policy evaluation appeal. An intriguing finding of the paper is that governance deficiency does correlate in aggregate with hotspot prevalence, despite many hotspot countries having poor governance in the overall rankings. This shifts the conversation from mining acceptability risks being considered as a developing world problem to one which has much broader context. This paper merits publication but there are a few areas which need revision or nuanced coverage before moving forward.	We have revised the manuscript according to these suggestions.
13	a) Demand forecast: The data in Figure 1 b can be questioned in terms of how metals like aluminum are relegated to a lower priority. Indeed a new report on critical metals for carbon mitigation from the World Bank (May, 2020) notes the salience of aluminum. The authors need to revisit the demand forecasts and consider this latest analysis with greater care:	We have revisited the metal demand forecast and produced a new figure. For this figure, we complemented our reference list with a review from Watari et al. (2020), bringing our number of included studies up to 17, and including the World Bank publication.

	http://pubdocs.worldbank.org/en/961711588875536384/Minerals-for-Climate-Action-The-Mineral-Intensity-of-the-Clean-Energy-Transition.pdf	
14	b) Static and dynamic risk categories: There is a major difference between some of the ecological risk categories and the social and governance risk categories in terms of temporal factors. The Amazon rainforest or the Congo basin has inherent biodiversity value and irreparable damage there suggests that this is a static risk which can't be diminished (hence call for permanent protected areas in some cases). On the other hand social and governance risks are dynamic and change frequently. Countries which may be stable one day can be unstable the next or vice versa. The authors need to differentiate these factors in their discussion – particularly with reference to their headline finding on platinum group metals. Given the dominance of South Africa in this regard and the dominance of Social and Governance factors in this conclusion, there should be a note about this matter.	We have amended the findings section (paragraph describing the ESG dimensions) to reflect this aspect.
15	Furthermore, there is now much higher recycling of platinum from catalytic converters as a result of such risks and authors need to note this aspect as well. The authors quote Rasmussen (2019) for their demand calculations but should note the recycling aspects in that paper as well.	Although recycling is part of the solution, this paper focuses on mining and mining-related risks. The study from Rasmussen anticipates high recycling rates from the fuel cell, autocatalyst and jewellery sectors in their 2050 forecast of global platinum demand. Total recycling rate is around 49%. In spite of this high recycling rate, 51% of platinum demand will still need to be met by the mining sector. We take this percentage into account when building figures 1c and 1d.
16	c) Processing impact: Surprisingly, rare earth metals come out as rather “green” in this analysis possibly because they are mined as companion metals in the larger deposits or the database does not cover the smaller mining sites in Jiangxi province of China. Furthermore, the chemical processing of rare earths has major environmental and social consequences, which are also not addressed in this analysis. Some discussion of this anomaly is worth addressing. The work of Dr. Julie Klinger in this arena may be worth consulting and referencing.	Rare earths come out with a balanced ESG risk profile because of their geographic location. Top countries with declared rare earths resources in our dataset are China, Canada, Russia, Vietnam, Japan, Greenland and Australia. It is worth noting however that the S&P database is less complete for rare earths than for other metals, as we show in table S1. We have added a reference to this table in the main text, as well as a comment underneath table S1 for added clarity. Also noting that technical considerations on mining and mineral processing steps, and how they differ from one metal to another, are out of the scope of this paper, which focusses on the external context in which mining activities take place. The ESG factors we define in this paper are relevant regardless of the commodity extracted.
17	d) Oceanic sources: Given the conclusion of this study that the technology metals, particularly cobalt will likely	We provided additional information on the Clarion-Clipperton seabed mining zone and its

	face major ESG risk, the authors should at least mention that oceanic minerals, particularly the massive reserves of cobalt deserve to be noted more clearly than a tangential reference in citation 27. This is particularly salient since the International Seabed Authority is currently formulating regulations precisely to address these concerns and provide alternative supply scenarios.	cobalt content, which, according to reference 27, is more than the entire global terrestrial reserve base.
18	Once these revisions are addressed, this paper would be ready for publication and a welcome contribution to the field.	Thank you

REVIEWERS' COMMENTS:

Reviewer #1 (Remarks to the Author):

The revised manuscript and response to the reviews have satisfactorily addressed my previous concerns. I recommend that it be accepted for publication.

Reviewer #2 (Remarks to the Author):

The revisions and response to reviewers are adequate and meet my expectations for publications.